# Higher Synovial Immunohistochemistry Reactivity of IL-17A, Dkk1, and TGF-β1 in Patients with Early Psoriatic Arthritis and Rheumatoid Arthritis Could Predict the Use of Biologics

**DOI:** 10.3390/biomedicines12040815

**Published:** 2024-04-08

**Authors:** Jose A. Pinto-Tasende, Mercedes Fernandez-Moreno, Ignacio Rego Perez, J. Carlos Fernandez-Lopez, Natividad Oreiro-Villar, F. Javier De Toro Santos, Francisco J. Blanco-García

**Affiliations:** 1Department of Rheumatology, Institute of Biomedical Research of A Coruña (INIBIC), Complexo Hospitalario Universitario de A Coruña, Universidade de A Coruña, 15006 A Coruña, Spain; jesus.carlos.fernandez.lopez@sergas.es (J.C.F.-L.); natividad.oreiro.villar@sergas.es (N.O.-V.); francisco.javier.toro.santos@sergas.es (F.J.D.T.S.); fblagar@sergas.es (F.J.B.-G.); 2Institute of Biomedical Research of A Coruña (INIBIC), Complexo Hospitalario Universitario de A Coruña, 15006 A Coruña, Spain; mercedes.fernandez.moreno@sergas.es (M.F.-M.); ignacio.rego.perez@sergas.es (I.R.P.)

**Keywords:** IL-17A, TGF-β1, Dkk1, IHC reactivity, psoriatic arthritis, synovial tissue, biomarker

## Abstract

Background: Delay in diagnosis and therapy in patients with arthritis commonly leads to progressive articular damage. The study aimed to investigate the immunohistochemical reactivity of synovial cytokines associated with inflammation and the bone erosives/neoformatives processes among individuals diagnosed with psoriatic arthritis (PsA), rheumatoid arthritis (RA), osteoarthritis (OA), and radiographic axial spondyloarthritis (r-axSpA), with the intention of identifying potential biomarkers. Methods: Specimens were collected from the inflamed knee joints of patients referred for arthroscopic procedures, and the synovial tissue (ST) was prepared for quantifying protein expression through immunohistochemical analysis (% expressed in Ratio_Area-Intensity) for TGF-β1, IL-17A, Dkk1, BMP2, BMP4, and Wnt5b. The collected data underwent thorough analysis and examination of their predictive capabilities utilising receiver operating characteristic (ROC) curves. Results: Valid synovial tissue samples were acquired from 40 patients for IHC quantification analysis. Initially, these patients had not undergone treatment with biologics. However, after 5 years, 4 out of 13 patients diagnosed with PsA and two out of nine patients diagnosed with RA had commenced biologic treatments. Individuals with early PsA who received subsequent biologic treatment exhibited significantly elevated IHC reactivity in ST for TGF-β1 (*p* = 0.015). Additionally, patients with both PsA and RA who underwent biologic therapy displayed increased IHC reactivity for IL-17A (*p* = 0.016), TGF-β1 (*p* = 0.009), and Dkk1 (*p* = 0.042). ROC curve analysis of IHC reactivity for TGF-β1, Dkk1, and IL-17A in the synovial seems to predict future treatment with biologics in the next 5 years with the area under the curve (AUC) of a combined sum of the three values: AUC: 0.828 (95% CI: 0.689–0.968; *p* 0.005) S 75% E 84.4%. Conclusions: Higher synovial immunohistochemistry reactivity of IL-17A, Dkk1, and TGF-β1 in patients with early psoriatic arthritis and rheumatoid arthritis may serve as potential indicators for predicting the necessity of utilising biologic treatments.

## 1. Introduction

Rheumatoid arthritis (RA) is the most common of the chronic systemic autoimmune diseases and is associated with increased disability and mortality [1]. Psoriatic arthritis (PsA) is a chronic inflammatory disease that affects the joints and skin of patients with psoriasis [2]. In many individuals with PsA, the disease exhibits characteristics of progressive and destructive changes. Significant pathological changes already occur in the early stages of PsA, and approximately half of the patients develop structural damage within 2 years of disease onset [3].

Both are associated with genetic, environmental, and immunological factors and the early diagnosis and treatment of RA and PsA can prevent disease progression and therefore prevent irreversible joint damage and disability [4,5].

RA and PsA exhibit distinct variations in clinical manifestation, radiographic observations, associated medical conditions, and underlying mechanisms, enabling differentiation between these prevalent types of chronic inflammatory arthritis [6]. Bone erosions lacking new bone formation and cervical spine engagement represent typical features of rheumatoid arthritis (RA). Conversely, axial spine involvement, psoriasis, and nail dystrophy are distinctive traits of psoriatic arthritis (PsA). Furthermore, individuals with PsA typically test negative for rheumatoid factor (RF) and antibodies to cyclic citrullinated peptide (CCP), whereas approximately 80% of those with RA exhibit positivity for RF and anti-CCP antibodies.

Although there is an overlap in the pathogenesis of PsA and RA, there are also differences that affect treatment efficacy. In PsA, levels of interleukin (IL)-1β, IL-6, IL-17, IL-22, IL-23, interferon-γ, and tumour necrosis factor-α (TNF-α) are elevated, while in RA, levels of IL-1, IL-6, IL-17, IL-22, IL-33, and TNF-α are elevated [7,8]. In both diseases, activation and invasion of T-cells and macrophages are considered essential in the initiation of inflammatory and destructive processes in the joints [9]. In RA, M1 macrophages (pro-inflammatory phenotype) become prevalent, leading to the secretion of high levels of pro-inflammatory cytokines, the activation of T and B cells through antigen presentation, and stimulation of bone resorption. In contrast, there is no difference in M2 (anti-inflammatory phenotype) cytokine expression between PsA and RA, but PsA is characterised by lower levels of M1 cytokine expression than RA [9,10].

The interplay among different immune cell types triggers the secretion of diverse cytokines, including IL-23, TNF-α, IL-17, and IL-22, and these cytokines play pivotal roles in fostering inflammation and activating resident cells within joint and enthesis tissues [11]. These cells, which include fibroblast-like synoviocytes, chondrocytes, osteoblasts, and osteoclasts, cause cartilage degradation, bone erosion, and joint destruction [12].

IL-17A is a pro-inflammatory effector cytokine produced by T helper (Th)17 cells, macrophages, mast cells, dendritic cells, natural killer cells, gamma/delta T cells, and CD8+ T cells [6,13]. Macrophages typically produce TGF-β1 during the clearance of apoptotic cells, aiding in the mitigation of inflammatory responses linked with phagocytosis. This cytokine plays a significant role in preserving tolerance by regulating the survival, proliferation, and differentiation of Th17 lymphocytes [14]. It is noteworthy that Th17 cells possess the capacity to serve both immunoregulatory and pathogenic functions [15].

TGF-β1 is a multifaceted cytokine participating in many biological processes like inflammation, fibrosis, lymphocyte recruitment, and lymphocyte–effector differentiation (Treg and Th17). In relation to this, Celis et al. [16] found a lower expression of TGF-β1 in the synovial tissue (ST) of PsA patients.

Previous studies in RA [17] and PsA [16] patients suggested that ST from an active joint is a good representation of other active joints in the same patient. We have previously reported [18] that IL-17A gene expression in the synovial membrane of patients with psoriatic arthritis was positively correlated with traditional proteins that damage the bone and negatively correlated with the bone-forming proteins in peripheral arthritis, and that TGF-β1 immunoreactivity in synovial tissue was higher in patients with erosive psoriatic arthritis correlating with the increased levels of IL-17A and Dkk1 in the IHC.

The present research aims to analyse the early cytokine profile concerning inflammation and bone destruction/regeneration in the ST of patients with psoriatic arthritis (PsA), comparing it with rheumatoid arthritis (RA), osteoarthritis (OA), and ankylosing spondylitis (AS), studying their role as possible biomarkers of worsening evolution and the need to initiate biological treatment in patients with arthritis.

## 2. Materials and Methods

The present study is a prospective follow-up of patients described in a previously published paper [18], which included patients fulfilling the CASPAR criteria for psoriatic arthritis (n = 13), RA (n = 9), OA (n = 18), and AS (n = 4). Between 2009 and 2013, the patients underwent knee arthroscopy due to clinical manifestations of knee joint swelling and tenderness that persisted despite receiving appropriate medical treatment for their diagnosed condition. Specimens were collected from the swollen knees of patients referred for arthroscopic procedures, and the synovial membrane was subjected to pathological examination and protein immunohistochemical quantification (% expressed in Ratio_Area-Intensity) for TGF-β1, IL-17A, Dkk1, BMP2, BMP4, and Wnt5b.

Patients diagnosed with rheumatoid arthritis and psoriatic arthritis were initiated on methotrexate therapy, with doses escalated up to 20 mg per week if well tolerated. In cases where no favourable response was observed or adverse effects emerged, patients were transitioned to treatment with TNF inhibitors or a combination therapy approach, according to treatment guidelines at that time for RA [19] and PsA [20]. Clinical and biological information, including tender and swollen joint counts (66/68), levels of C-reactive protein (CRP), and erythrocyte sedimentation rate (ESR), alongside details of disease-modifying antirheumatic drugs (DMARDs) and biologic therapies administered, were gathered at both the study’s commencement and the final clinical assessment. Additionally, the degree of osteoarthritis in the knee was determined using the Kellgren–Lawrence (KL) scale [21]. Assessment of axial radiological damage included sacroiliac X-ray examinations and classification of sacroiliitis based on the modified New York criteria [22] as well as to the presence/absence of erosive peripheral joint damage. This study was carried out at the A Coruña Biomedical Research Institute (INIBIC) and had been approved by the Clinical Research Ethics Committee (CEIC) under the reference number 2011/301 and conducted following the principles outlined in the Declaration of Helsinki. Written informed consent was obtained from all participating patients before their inclusion in the study.

### 2.1. Arthroscopies

The arthroscopies were performed as described elsewhere [18] and ST samples were obtained surgically from the knee joint using a 2.7 mm arthroscope (Storz, Tullingen, Germany) under local anaesthesia, and tissue samples were immediately fixed in 4% formaldehyde. These samples were then embedded in paraffin wax for subsequent immunohistochemical analysis [16]. Biopsies from each patient were promptly collected, prepared, and fixed in the operating room. Alongside blood samples, they were swiftly transferred to the INIBIC facilities within one hour. Upon arrival in the laboratory, the samples were logged into the database (Biobank, Área Sanitaria A Coruña e Cee, Complexo Hospitalario Universitario A Coruña (CHUAC), A Coruña, Spain) using NorayBanks software (V3.60.2310.1015-NorayBanks2022) to ensure procedural confidentiality. Following encoding, they underwent processing according to the described techniques.

### 2.2. Histopathological Analysis and Immunohistochemistry: Quantification of Protein Expression in IHC Staining

Synovial biopsies obtained from each patient were promptly fixed in the operating room. Some biopsies were immersed in 4% formaldehyde for a maximum of 24 h, while others were placed in OCT (cryoprotective medium). Following fixation, they were transported in dry ice for storage, adhering to a predetermined order, in a −80 °C freezer located in the Basic Research Laboratory at the INIBIC. Once the collection phase was completed, sections were prepared and stained with haematoxylin–eosin and Masson’s trichrome (H&E, MM-classical histological stains) for initial morphological examination. Indirect immunohistochemistry techniques (with peroxidase in paraffin) were used on all biopsies, using as primary antibodies the mouse monoclonal anti-BMP-2 antibody ab6285 (clone 65529. 111) from Abcam^®^ (Abcam: Discovery Drive, Cambridge, Biomedical Campus, Cambridge, UK), the rabbit anti-BMP-4 monoclonal antibody ab39973, the mouse anti-Wnt5b monoclonal antibody ab86720 (clone 3D10) from Abcam^®^, the rabbit anti-DKK1 monoclonal antibody ab109416 (clone EPR4759) from Abcam^®^, the anti-TGF-β1 antibody ab64715 (clone 2Ar2) from Abcam^®^, and the rabbit anti IL-17 polyclonal antibody ab79056 from Abcam^®^. Dako^®^ K-5007 antibody (EnVision™ Detection Systems Peroxidase/DAB, 70 Saint Vincent St., Nelson, New Zealand) was used as a secondary antibody. The samples underwent pretreatment with tris-EDTA at pH 9 using a Retriever (or 0.1 M sodium citrate at pH 6.1 for Wnt5b). Positive controls included BMP2 for human small bowel tissue (dilution: 1:5000), BMP4 for human colon tissue (dilution: 1:1000), Dkk1 for human placenta tissue (dilution: 1:1000), Wnt5b for human thyroid tissue (dilution: 1:1000), TGF-β1 for human articular cartilage tissue and osteoarthritis (dilution: 1:50), and IL-17A for human lymphatic node (dilution: 1:1000). Negative controls were conducted without the use of a primary antibody. The chromogen utilised in this process was diaminobenzidine (DAB), which imparts a brown colour, along with the peroxidase substrate (H_2_O_2_). Following immunohistochemical staining, the samples were counterstained with Gill III’s haematoxylin–eosin, subsequently dehydrated, and rinsed with xylene. DPX (acrylic resin) was used as a coverslip mounting medium. A fluorescence confocal microscope was used for semi-quantitative measurements of the synovial tissue. Image capture was carried out using the Olympus BX61 microscope (Shinjuku-ku, Tokyo, Japan), while analysis and quantification of the samples were conducted with the Nikon Eclipse microscope (Stroombaan 14, Amstelveen, The Netherlands), utilising the NIS Elements imaging software (https://www.nikon.com/).

### 2.3. Statistical Analysis

Descriptive statistics for continuous variables were presented as percentages or as the median and interquartile range (IQR), while categorical variables were expressed as frequencies and percentages. Comparative analyses of qualitative variables utilised the chi-squared test, with Fisher’s exact test employed where applicable. The Wilcoxon rank sum test or Kruskal–Wallis’s test was employed to compare the distribution of numeric variables among groups. For correlation between two categorical variables or between one numeric and one categorical variable, Fisher’s exact test and the Wilcoxon rank sum test or Kruskal–Wallis’s test were utilised. Pairwise statistical significance was obtained by the Dunn–Bonferroni test when the Kruskal–Wallis test was calculated.

Univariate and multivariate logistic regression models were employed to assess the association of proteins related to IHC expression with demographics, clinical characteristics, radiological findings, and therapeutic interventions. Data from the study of protein IHC quantification (% expressed in Ratio_Area-Intensity) for TGF-β1, IL-17A, Dkk1, BMP2, BMP4, and Wnt5b were used to compare patients who had started biologics with those who had not. Receiver operating characteristic (ROC) curves were used to analyse the ability of cytokines’ profile to predict the worst evolution and need to start biologics. Sensitivity, specificity, and Youden Index were calculated. Youden Index ranges between 0 and 1, with 0 values indicating that a diagnostic test gives the same proportion of positive results for groups with and without the disease. Values of 1 indicate that there are no false positives or false negatives.

The data were statistically analysed with the SPSS version 21 programme (IBM SPSS Statistics). Values of *p* < 0.05 were considered statistically significant.

## 3. Results

Synovial tissue samples were obtained from 44 patients (PsA n = 13, RA n = 9, OA n = 18, AS = 4). Forty were valid for IHC quantification analysis, and four were not valid for the study. Two patients with arthritis (one with PsA and one with RA) and two with osteoarthritis were excluded because their IHC readings of BMP4 and Wnt5b could not be performed correctly. Baseline features are described in Table 1. At baseline, eight patients were treated with MTX (53.8%), while none received biologic therapies. After a median follow-up of 5 years, 4 out of 13 patients with psoriatic arthritis (31%) and two out of nine patients with rheumatoid arthritis (22.2%) received biologic treatments. Overall, immunohistochemical reactivity for TGF-β1 in synovial tissue was elevated in patients with psoriatic arthritis (*p* = 0.024) and was positively correlated with IL-17A (r = 0.355, *p* = 0.024) and Dkk1 (r = 0.444, *p* = 0.004). In patients with PsA who received biologic therapy afterwards, there was a greater early immunohistochemical reactivity for TGF-β1 in the synovial tissue compared to those who did not receive biologics (*p* = 0.015).

Patients with PsA and RA treated with biologics had higher IHC reactivity in the early synovial tissue for TGF-β1 (*p* = 0.009), IL-17A (*p* = 0.016), and Dkk1 (*p* = 0.042) (Table 2 and Figure 1).

No significant differences were observed in terms of gender (*p* = 0.944), age (*p* = 0.947), or Kellgren–Lawrence scale scores in the target knee (*p* = 0.608) concerning TGF-β1 expression. However, it was noted that IL-17A immunohistochemical expression was higher in men (*p* = 0.02). In the logistic regression analysis, it appears that the observed differences in TGF-β1 (*p* = 0.014), IL-17A (*p* = 0.009), and Dkk1 (*p* = 0.015) expressions are independent of age, gender, or conventional synthetic disease-modifying antirheumatic drug (csDMARD) use at baseline. Regarding TGF-β1, there were no significant differences observed concerning gender, age, or radiological involvement. However, for IL-17A, it was noted that immunohistochemical levels were higher in men (*p* = 0.013). The positive coefficient (B) of 2.059 suggests that male gender individuals are 7.840 times more likely to use biologicals compared to female gender individuals (*p* = 0.013), holding other variables constant (Appendix A).

Early IHC reactivity in patients with RA and PsA who received biologics was higher for DKK1, IL-17, and TGF-β1 levels (Figure 2), and then the study of the area under the curve was carried out considering these values with statistically significant differences. The analysis expressed as the area under the curve indicates that Dkk1, TGF-β1, and IL-17A levels have a sufficiently high sensitivity and specificity to predict the use of biologics in these patients.

The Receiver-operating characteristic curve analysis results of synovial tissue, cutoff value, and Youden’s Index were as follows (Figure 3): for IL-17A AUC: 0.805 (95% CI: 0.656–0.954; *p* 0.008) S 75% E 78.1%, cutoff value of IL17A IHC (%) = 18, and Youden’s Index: 0.53; for TGF-β1 AUC: 0.792 (95% CI: 0.636–0.948; *p* 0.011) S 75% E 69.7%, cutoff value of TGF-β1 IHC (%) = 7, and Youden´s Index: 0.45; for DKK1 AUC: 0.758 (95% CI: 0.580–0.936; *p* 0.026) S 75% E 68.7%, cutoff value of DKK1 IHC (%) = 40, and Youden’s Index: 0.44. For the combined sum of the 3 values with the same cutoffs: AUC: 0.828 (95% CI: 0.689–0.968; *p* 0.005) S 75% E 84.4%, and Youden´s Index: 0.59 (Figure 4).

The immunohistochemical expression of TGFβ1, IL-17A, and DKK1 in the synovial tissue of patients treated with and without biologics while affected by their disease can be found in Figure 5.

## 4. Discussion

Early treatment strategy is gradually improving the prognosis of patients with newly diagnosed RA or PsA, and the search for new prognostic biomarkers is becoming increasingly necessary [23,24,25,26,27]. Having a diagnostic tool to predict the evolution of these patients could be very useful [28]. To date, many studies have evaluated the ability of different parameters to predict poor prognosis [29,30,31]. Surrogate markers of disease severity, such as the need for biological agents or estimation of treatment intensity, could provide useful results [32]. Several groups of researchers have analysed treatment-associated variables, such as the group of Verstappen et al. [33], who reported that failure of the first DMARD is a marker for the need for biologic agents in patients with early inflammatory polyarthritis.

Biomarkers have been studied to determine whether they can predict the need for intensive treatment in patients with early arthritis. Several papers discuss the value of biomarkers in this regard. González-Alvaro et al. [32] suggest that traditional and new biomarkers can help to adjust treatment to disease activity or poor outcomes, but, so far, no biomarker can bridge the gap between disease onset and prescription of the first DMARD. Teitsma et al. [34] developed a model based on protein biomarkers that showed an additional predictive value to clinical predictors in determining the need for biological therapy. Da Mota et al. [35] found that higher levels of rheumatoid factor (RF) and anti-CCP over time were associated with the need for biologic therapy in RA. Nam et al. [36] discuss the role of biomarkers in assessing disease severity and monitoring disease activity for individualised therapy.

Since the synovial membrane is the focus of inflammation in RA and PsA (although in this case, the enthesis is as or more important than the synovial membrane), it is necessary to directly analyse cytokines in this tissue, and when analysed by immunohistochemistry, more cells in RA and PsA synovial tissue express pro-inflammatory cytokines [37].

The pathophysiology of RA involves all cells responsible for the innate and adaptive immune response and resident cells in the synovial membrane. Thus, the synovial membrane is transformed into a growing mass of tissue or pannus that can overlap and invade the surface of cartilage and bone. This pannus behaves like a locally invasive tumour whose components, macrophages, osteoclasts, and fibroblast-like synoviocytes degrade bone and cartilage [38].

An analysis of the synovial membrane of PsA has shown the presence of macrophages in the lining layer, while different types of immune cells (macrophages, mast cells, polymorphonuclear cells, T cells, B cells, and plasma cells), which produce a wide range of cytokines, can be found in the sub-layer [11].

Our data show that patients with RA and PsA who required more therapeutic efforts comprising the use of biologic therapy during follow-up had higher expressions of IL17A, TGFb1, and Dkk1 in the synovial tissue of inflamed knees compared to patients who did not require bDMARD treatment, as well as to patients with PsA and RA, and the control group (osteoarthritis and axial spondyloarthritis).

Studies have shown that TGF beta levels are elevated in the synovial fluid and sera of RA patients, indicating its potential as a biomarker of disease. TGF beta is associated with disease activity and severity in RA, and higher levels correlate with more severe symptoms and joint damage [39,40,41,42]. TGF beta has also been implicated in the regulation of immune responses in RA and PsA, suggesting its potential as a biomarker for monitoring treatment response and disease progression. In line with the above, we have observed in our study an increased early IHC reactivity for TGF-β1 in synovial tissue in PAs treated after biologics compared to those left untreated.

IL-17A is the hallmark of the Th17 subset of CD4 lymphocytes, which also produce IL-17F, IL-22, and IL-21 [43]. Th17 cell differentiation relies on the presence of pro-inflammatory cytokines such as IL-6 and IL-1β. However, surprisingly, TGFβ1, typically produced by various non-immune cells, also plays a pivotal role in this process. Experimental animal models of chronic inflammatory conditions such as posterior uveitis, psoriasis, inflammatory bowel disease, multiple sclerosis, collagen-induced arthritis, and ankylosing spondylitis have highlighted the significant pathogenic impact of IL-17A [44]. However, IL17A blockade is very effective only for psoriasis, psoriatic arthritis, and axial SpA, and several monoclonal antibodies that selectively target this cytokine have now been approved and marketed.

IL-17A+ CD8+ T cells have been identified in the synovial tissue of psoriatic arthritis patients and have been shown to induce pro-inflammatory responses in synovial fibroblasts [45]. In addition, IL-17Ai treatment with secukinumab has been shown to modulate the expression of genes related to immune and inflammatory responses, including those involved in bone remodelling, in the synovium of patients with spondyloarthritis and psoriatic arthritis [46]. These findings suggest that IL-17A in synovial tissue may have potential as a biomarker for psoriatic arthritis.

Our results show that IHC reactivity in patients with early RA and PsA, who subsequently received biologics for their more severe course, was higher for IL-17A levels in synovial tissue, so it could be a biomarker of worse outcomes. Action on the IL-17 axis is effective in psoriasis and spondyloarthritis, but not in rheumatoid arthritis. However, the expression and functional response to IL-17A appears to be similar in the synovitis of psoriatic arthritis and rheumatoid arthritis [47]. Dkk-1 is abundantly expressed in inflamed joints of destructive and remodelling forms of arthritis, playing an important role in the pathogenesis of RA and PsA [48,49].

Various inflammatory joint diseases manifest distinct patterns of bone damage. Rheumatoid arthritis typically presents with pronounced erosions. In contrast, psoriatic arthritis often displays a combination of bone destruction and formation, and Axial SpA is characterised by new bone formation [49]. Elevated serum levels of Dkk1 have been observed in patients with PsA and RA, and these levels are associated with disease severity and activity [50]. Dkk1 is the main suppressor of the Wnt signalling pathway, leading to decreased osteoblast proliferation, and contributing to bone erosion in PsA and RA [51,52]. The expression of Dkk1 appears to play a critical role in determining the trajectory of joint remodelling in arthritis. When Dkk1 is overexpressed, it tends to shift the process towards an erosive and destructive phenotype. Conversely, decreased expression of Dkk1 is associated with new bone formation within the affected joint [53,54]. Abnormal elevation of Dkk1 in patients with PsA and RA is thought to be involved in the process of structural radiographic alterations and the development of bone erosion [55,56]. In addition, Dkk1 has been identified as a possible diagnostic marker for early PsA, and its levels are higher in early PsA compared to established PsA [57].

The ROC curve is an analytical method, represented as a graph, used to evaluate the performance of a binary diagnostic classification method of a test and to select an optimal cutoff value for determining the presence or absence of a patient characteristic. The AUC is an effective combined measure of sensitivity and specificity that describes the inherent validity of diagnostic tests. In the present study, non-parametric ROC curves were plotted for IHC expression in the TS of three cytokines separately (Figure 3) and for a combination of the three (Figure 4), using a non-parametric method. This curve and the corresponding AUC show that the combination of the three cytokines (TGF-β1, Dkk1, and IL-17A) as biomarkers have the predictive capacity to discriminate those patients with more severe arthritis evolution and, therefore, who will require the use of more forceful, but also more costly, therapeutic measures. In addition, the Youden Index has been used, which uses the maximum of the vertical distance of the ROC curve from the point (x, y) on the diagonal line (line of chance). In fact, the Youden Index maximises sensitivity + specificity between several cutoff points, allowing us to calculate the optimal cutoff point.

In the present study, our results showed that the synovial tissue’s IHC expression of IL-17A, Dkk1, and TGF-β1 correlated with each other. Furthermore, analysis of the data obtained in our study, expressed as the area under the curve, indicates that the levels of Dkk1, TGF-β1, and IL-17A have a sufficiently high sensitivity and specificity to reasonably predict the use of biologics in these patients within 5 years of the onset of arthritis.

We followed these patients to see if any of the cytokines analysed could differentiate those patients with a milder clinical course of arthritis from those with a more moderate or severe evolution and who, therefore, required a more intense or aggressive therapeutic intervention.

Therefore, so far, no biomarker can tell us from the early stages of the disease which patients will progress worse and require more intensive treatment from the beginning and will need a prescription of csDMARDs or the first bDMARD. Our results suggest that IHC expression of cytokines related to the inflammatory process in inflamed RA synovial tissue and PAs in early stages of the disease could serve as a proxy in this regard.

Although there is a suggestion that therapeutic decisions in psoriatic arthritis (PsA) should be customised based on the unique characteristics of each patient, the treatment of psoriatic disease still falls short of achieving precision medicine [58,59]. Indeed, there is a lack of validated biomarkers capable of accurately predicting responses to specific therapies [60], and the selection of medication is primarily influenced by various factors, including disease severity, the presence of extra-articular manifestations, disease endotypes, prognostic factors, previous treatment history, comorbidities, access to therapy, and patient preferences. This decision-making process closely resembles that of other related diseases such as rheumatoid arthritis [61,62].

Our study has several important limitations: its single-centre, open-label design; its small sample size; and the fact that synovial biopsy is an invasive method that is not without risk and requires an arthroscope and an operating theatre for its use. ST samples could be obtained from a relatively limited number of patients and utilised to quantify biomarkers associated with inflammation, joint degradation, and pharmacological effects. To mitigate inter-patient variability and bolster the statistical robustness of the study, it is recommended to assess serial synovial biopsy samples. Typically, around 10 patients per group are deemed sufficient for this type of investigation, although the exact number may vary depending on the specific biomarker being analysed [37]. These results should be treated with caution until they can be verified in future studies with larger sample sizes.

## 5. Conclusions

Patients treated with biologics showed increased IL-17A, TGF-β1, and Dkk1 IHC reactivity at baseline. These differences were independent of age, gender, and treatment with csDMARDs. In summary, these data suggest that TGF-β1, Dkk1, and IL-17A IHC in PsA and RA patients could anticipate the need for biological therapies, but these findings need to be approached carefully until they are confirmed by larger studies in the future.

## Figures and Tables

**Figure 1 biomedicines-12-00815-f001:**
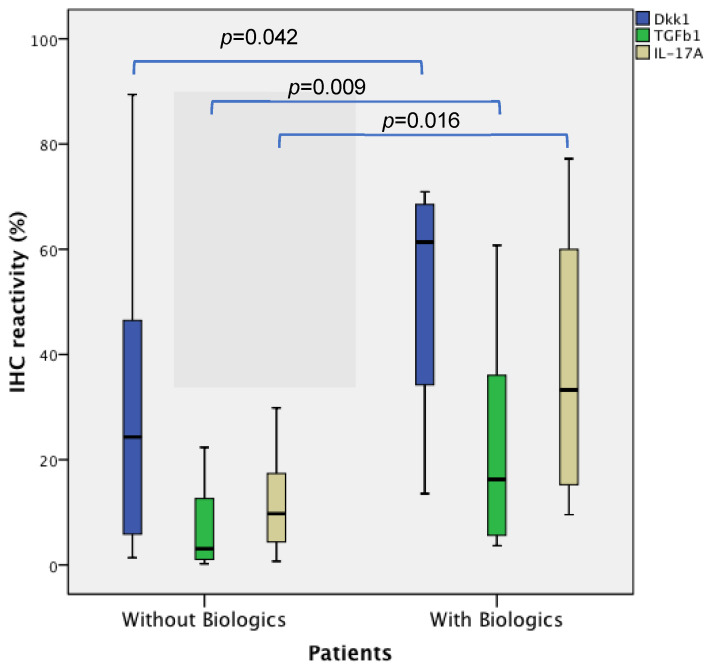
Differences in baseline IHC expression levels of Dkk1, TGFβ1, and IL-17A in the synovial tissue, whether biologics were required or not. Synovial tissue IHC expression of the 3 cytokines (IL-17A, TGF-β1, and Dkk1) in all patients (PsA, RA, OA, and r-axSpA) is shown divided into 2 groups: those who have received biological treatment and those who have not received biological treatment.

**Figure 2 biomedicines-12-00815-f002:**
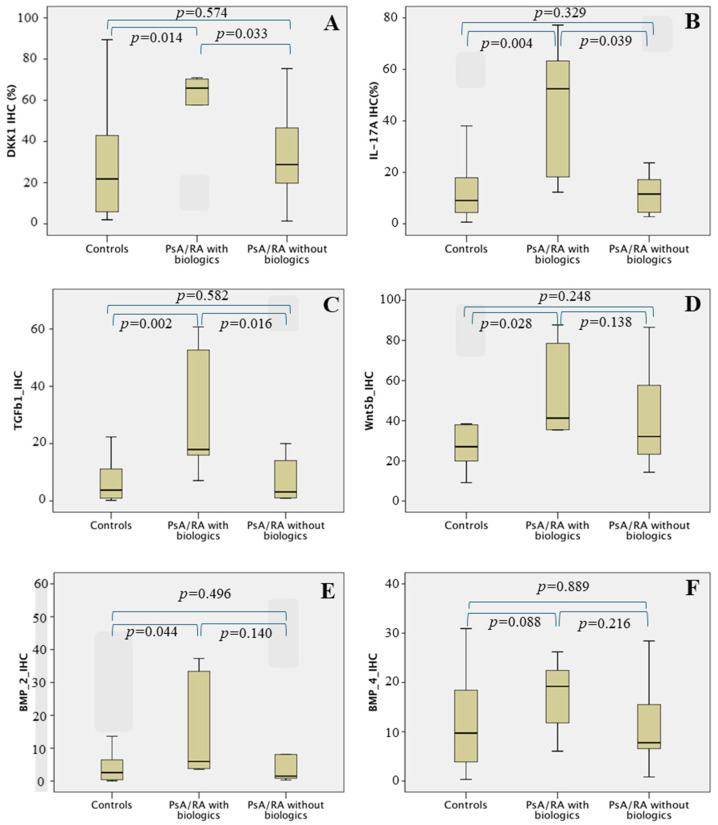
Boxplots of IHC reactivity in the early synovial tissue in the 3 groups: Controls, PsA/RA treated with biologics, and PsA/RA patients treated without biologics. Controls: osteoarthritis and axial spondylarthritis patients. The y scale represents the relative intensity percentage of IHC reactivity. Numbers in the graphs represent the identification code of patients with extreme IHC quantification values. Pairwise statistical significance was obtained by the Dunn–Bonferroni test when the Kruskal–Wallis test was calculated. The IHC expression ratio is significantly higher in the PsA/RA group treated with biologics for Dkk1 (**A**), IL-17A (**B**), and TGF-β1 (**C**) compared to the control group and the PsA/RA group treated without biologics. In the case of Wnt5b (**D**), BMP2 (**E**), and BMP4 (**F**), IHC expression is also higher in the PsA/RA group treated with biologics compared to the control group but does not reach statistical significance in the case of BMP4.

**Figure 3 biomedicines-12-00815-f003:**
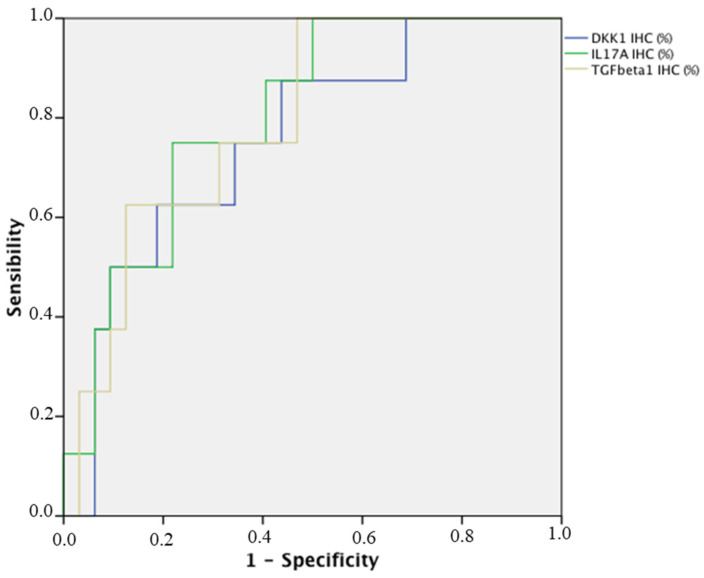
ROC curve analysis of IHC reactivity for TGF-β1, Dkk1, and IL-17A in the synovial tissue. ROC: Receiver-operating characteristic curve analysis of synovial tissue.

**Figure 4 biomedicines-12-00815-f004:**
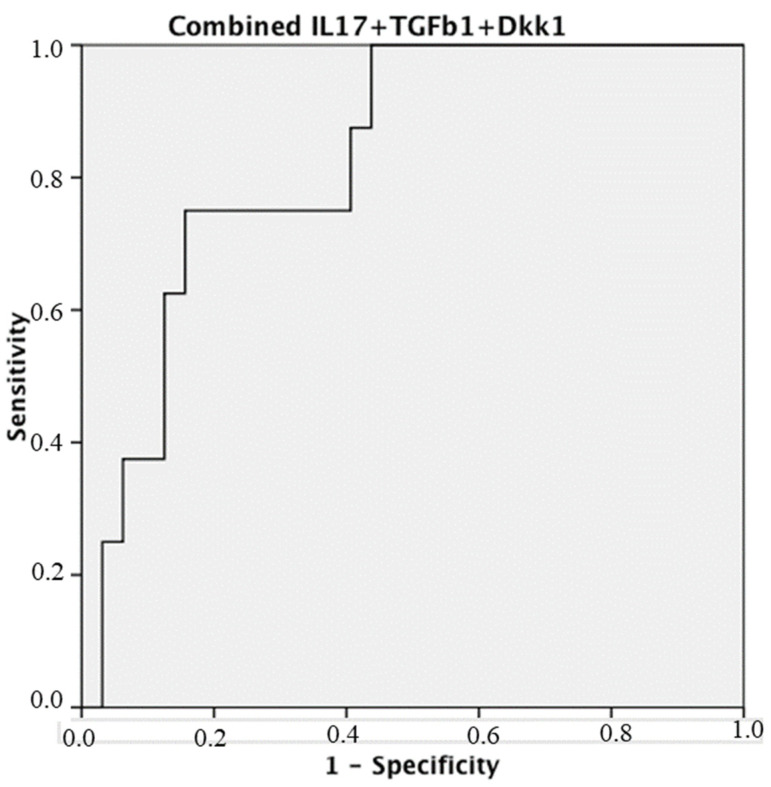
ROC curve analysis of IHC reactivity combined sum for TGF-β1, Dkk1, and IL-17A in the synovial tissue. ROC: Receiver-operating characteristic curve analysis of synovial tissue. Score with a combined sum of the 3 values: AUC: 0.828 (95% CI: 0.689–0.968; *p* = 0.005) S 75% E 84.4%. Cutoff value of IHC combined sum = 90. Youden’s Index: 0.59.

**Figure 5 biomedicines-12-00815-f005:**
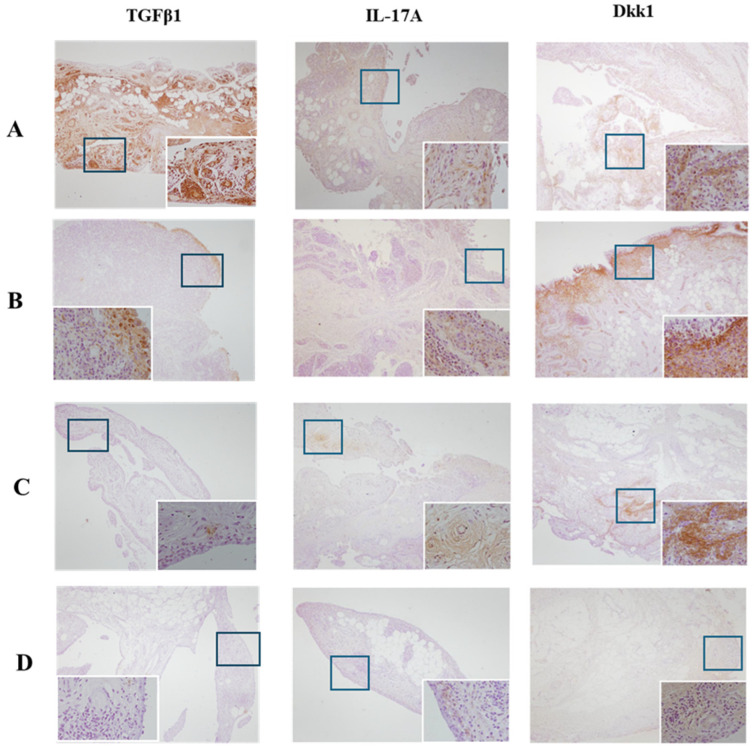
Immunohistochemical expression of TGFβ1, IL-17A, and DKK1 in the synovial tissue of patients treated with and without biologics while affected by their disease. Representative images of synovial tissue of swollen knee from each group of study. Synovium sections were analysed by immunohistochemistry (brown) for TGFβ1, IL-17A, and Dkk1, expressed in the synovial from: (**A**) PsA treated with biologics; (**B**) RA treated with biologics; (**C**) PsA treated without biologics; (**D**) Controls (OA/axSpA). All tissues were counterstained with haematoxylin from each group of study (original magnification 20×, 100×).

**Table 1 biomedicines-12-00815-t001:** Baseline characteristics of patients according to diseases.

Total n = 44 *	PsA n = 13	RA n = 9	OA n = 18	r-axSpA n = 4	*p* **
Male, n (%)	10 (66.7)	4 (50.0)	7 (30.9)	4 (100)	0.036
Age, years	48.0 (40.5–55.5)	42.5 (32.2–58.7)	69.0 (64.5–74.7)	57.5 (49.0–65.2)	<0.0001
Time course of the disease, years	3.0 (1.0–9.0)	2.0 (1.0–5.7)	12.5 (6.0–18.5)	19.5 (8.0–31.7)	0.002
Tender joint count	2 (1–3)	1 (1–2)	1 (1–1)	1 (1–1.7)	0.033
Swollen joint count	1 (1–2)	1 (1–1.7)	1 (1–1)	1 (1–1)	0.236
ESR, mm/first hour	20 (8–36)	15 (7–29)	22 (14–33)	45 (23–69)	0.181
C-reactive protein, mg/dL	0.70 (0.29–1.74)	0.36 (0.21–0.61)	0.44 (0.25–0.53)	2.20 (0.52–3.80)	0.05
Uricaemia, mg/dL	4.8 (4.1–6.3)	5.7 (4.2–7.7)	5.5 (4.5–6.0)	5.2 (4.2–6.9)	0.715
RF and/or ACPA+, n (%)	0	4 (50)	0	0	0.001
Erosions on X-rays, n (%)	4 (26.7)	0	0	0	0.032
Sacroiliitis on X-rays, n (%)	0	0	0	4 (100%)	0.001
Metotrexate, n (%)	8 (53.3)	0	0	0	0.004

* Data as medians (Q1–Q3) and percentages. ** Chi-squared/Fisher’s exact test) and Kruskal–Wallis test. RF: rheumatoid factor; ACPA: anti-citrullinated peptide antibody; ESR: erythrocyte sedimentation rate. HLA-B27+: four patients with radiographic axial spondyloarthritis (r-axSpA).

**Table 2 biomedicines-12-00815-t002:** IHC reactivity in the early synovial tissue in PsA and RA patients treated with biologics or not.

IHC *	Controls n = 20	PsA/RA with Biologic n = 6	PsA/RA without Biologics n = 14	*p* **
**DKK1**	21.82 (5.36–43.37)	61.37 (30.51–69.40)	24.31 (5.36–46.54)	**0.042**
**BMP2**	2.57 (0.38–7.29)	4.18 (3.62–26.98)	1.82 (0.59–8.13)	0.127
**BMP4**	9.72 (3.44–18.91)	16.54 (10.32–21.71)	8.00 (4.81–18.91)	0.260
**Wnt5b**	27.06 (18.80–38.16)	36.32 (26.07–70.28)	30.81 (22.53–50.58)	0.067
**TGFβ1**	3.82 (0.94–11.55)	16.24 (4.88–44.33)	3.06 (1.06–12.94)	**0.009**
**IL17A**	9.07 (4.30–18.09)	33.26 (13.78–61.63)	9.75 (4.31–17.47)	**0.016**

* Data as median (Q1–Q3) and percentages. ** Kruskal–Wallis’s test. The *p* < 0.05 obtained in the comparative analysis of the 3 groups have been considered. Controls: patients diagnosed with osteoarthritis and axial spondyloarthritis. Bold: statistical significance.

## Data Availability

The datasets generated and/or analysed during the current study are not publicly available due to privacy or ethical restrictions but are available from the corresponding author upon reasonable request.

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
