# Peer review of "Higher Synovial Immunohistochemistry Reactivity of IL-17A, Dkk1, and TGF-β1 in Patients with Early Psoriatic Arthritis and Rheumatoid Arthritis Could Predict the Use of Biologics"

_biomedicines, 2024, doi:10.3390/biomedicines12040815_

Round 1
Reviewer 1 Report
Comments and Suggestions for Authors
The authors evaluated in synovial biopsies of early psoriatic arthritis patients (PsA) and rheumatic arthritis (RA) several potential biomarkers (cytokines and bone remodeling factors) using quantitative immunohistochemistry (IHC). They report that high synovial IHC reactivity at the baseline could predict future use of biologic DMARDs.
General comment:
Overall, the study is very limited in scope and adds very little to the current knowledge. The main issue is the tiny size of the patient cohorts (13 PsA and 9 RA), with only 4 PsA and 2 RA patients receiving biologic DMARDs 5 five years following diagnosis, thereby questioning whether the sample sizes are sufficient to achieve adequate power.
While the study may be helpful, there are significant limitations for potential application in clinics.
Minor issues:
Abstract, line 29: There are two p values for Dkk1.
Page 2, line 73: Please include the gamma/delta T-cells as IL-17 producers.
Histopathological analysis: Please add further details on how IHC data were normalized. The IHC technique is semi-quantitative at best.
Figure 2: What represents the y scale?
Figures 1-4: More informative figure legends are needed.
Page 10, line 286: PsA
Page 10, line 312: IL17A+ CD8+ is in duplicate.
Author contributions: MEVM (M. E. Vazquez-Mosquera?) is not among the author list.
Comments on the Quality of English LanguageMinor editing issues.
Author Response
Reply to Reviewer 1
General comment:
Overall, the study is very limited in scope and adds very little to the current knowledge. The main issue is the tiny size of the patient cohorts (13 PsA and 9 RA), with only 4 PsA and 2 RA patients receiving biologic DMARDs 5 five years following diagnosis, thereby questioning whether the sample sizes are sufficient to achieve adequate power.
While the study may be helpful, there are significant limitations for potential application in clinics.
Reply: Thanks for your comments. A potential limitation of biomarker studies is the statistical power of the study design. The power of a study depends on several aspects, not least of which is the sample size. In the case of tissue-sampled biomarkers, tissue sampling is often technically difficult to obtain, or the tissue obtained is not viable, and published studies do not tend to reach large sample sizes. However, despite the limited sample size, we have been able to observe notable differences between the different diseases.
We appreciate and benefit from your comments. Although some inherent limitations cannot be resolved, we have tried our best to modify them and are eager for your understanding. Thanks again.
Minor issues:
Abstract, line 29: There are two p values for Dkk1.
Reply: Thanks for your suggestions. We have revised the inappropriate presentation in the manuscript (Page 1, line 29), and corrected the erroneous data in the abstract.
Page 2, line 73: Please include the gamma/delta T-cells as IL-17 producers.
Reply: Thanks for your suggestions. We have included gamma/delta T-cells as IL-17 producers.
Histopathological analysis: Please add further details on how IHC data were normalized. The IHC technique is semi-quantitative at best.
Reply: Thanks for your suggestions. The confocal microscope’s ability to block out-of-focus light and thereby perform optical sectioning through a specimen allows it to quantify fluorescence with very high spatial precision. We have added that a fluorescence confocal microscope was used for semi-quantitative measurements of the synovial tissue.
Figure 2: What represents the y scale?
Reply: Thanks for your suggestions. The y scale in Figure 2 represents the relative intensity percentage. This clarification is added at the bottom of Figure 2.
Figures 1-4: More informative figure legends are needed.
Reply: Thanks for your suggestions. We have improved and completed the figures and legends.
Page 10, line 286: PsA
Reply: Thanks for your suggestions. PAs has been changed to PsA.
Page 10, line 312: IL17A+ CD8+ is in duplicate.
Reply: Thanks for your suggestions. The duplicate has been removed.
Author contributions: MEVM (M. E. Vazquez-Mosquera?) is not among the author list.
Reply: Thanks for your suggestions. MEVM has been removed and IRP has been added to the Authors’ contributions list.
We appreciate and benefit from your comments. Although some inherent limitations cannot be resolved, we have tried our best to modify them and are eager for your understanding. Thanks again.
Reviewer 2 Report
Comments and Suggestions for Authors
The manuscript submitted by Pinto-Tasende and colleagues aims to highlight the predictive value of IL17A, Dkk1 and TGF-b1 immunohistochemistry reactivity for the application of biologics in RA and PsA patients. The authors have indicated the limitations of their study - small sample number and thus, limited significance of the results, use of an invasive method to obtain samples. A main question raised by the study is whether the serum levels of IL-17A, Dkk1 and TGF-b1 correspond to the detected increased early IHC reactivity in RA and PsA patients who received biologics.
Points for revision:
1. Abstract:
1.1. L22 - Il-23, IL-6, IL-22, Sclerostin, Wnt1 quantifications were done but not shown in the results section and were not discussed. Thus, they should be removed from the abstract.
1.2. L29: doubled Dkk1. Which is the correct p value for Dkk1?
2. Introduction:
2.1. L58-59: IL-17 levels were shown to be elevated in RA patients. Revise the cytokine list for RA patients.
2.2. Unify abbreviations: IL17A or IL-17A, T-cells (L61) or T cells.
2.3. L82: revise the beginning of the sentence.
2.4. L92: it should be written ST instead of TS.
3. Materials and methods:
3.1. L119: delete "study".
3.2. L139-140: revise the sentence.
4. Results:
4.1. L187-188: the sentence states that 9 patients were treated with MTX but Table 1 shows that 8 patients were treated with MTX?
4.2. L188-193: the described data is not presented with figure or table.
4.3. Table 1: it is reasonable to show in separate column characteristics of patients treated with biologics after 5 years.
4.4. Table 2: define the controls.
4.5. Figure 1: revise the figure caption (... synovial tissue of patients treated with biologics or not.); indicate Patients instead of "Biologics", Without biologics instead of "Not" and With biologics instead of "Yes"; add IHC (%) on the y axis; define the type of statistical test used. What do the values 73, 1, 28, 43 indicate?
4.6. Table 3: what is the data for female patients?
4.7. Figure 2: improve the quality of the figure; define the controls and the statistical test used.
4.8. L215, L 238: delete the comma after "constant" and "0.59".
5. Discussion:
5.1. the section should include comparison with other IHC evaluations in RA and PsA patients and their possible relation with serum levels of the measured signaling molecules.
5.2. L286, L301: PAs or PsA?
5.3. L312: Double IL17A+CD8+.
5.4. L345: IHQ or IHC?
6. Conclusions: L369: delete "after 5 years".
Comments on the Quality of English Language
The sentences that need revision are indicated in the section "Comments and suggestions for authors".
Author Response
Reply to Reviewer 2
Comments in general:
The manuscript submitted by Pinto-Tasende and colleagues aims to highlight the predictive value of IL17A, Dkk1 and TGF-b1 immunohistochemistry reactivity for the application of biologics in RA and PsA patients. The authors have indicated the limitations of their study - small sample number and thus, limited significance of the results, use of an invasive method to obtain samples. A main question raised by the study is whether the serum levels of IL-17A, Dkk1 and TGF-b1 correspond to the detected increased early IHC reactivity in RA and PsA patients who received biologics.
Reply: Thanks for your suggestions. This is an excellent question. We had performed ELISA determination of serum levels of the different cytokines involved in the study and when we analysed the correlation with IHC, the serum levels of IL-17A, Dkk1 and TGF-b1 did not correlated with the detected increase in IHC reactivity in the patients.
Points for revision:
- Abstract:
1.1. L22 - Il-23, IL-6, IL-22, Sclerostin, Wnt1 quantifications were done but not shown in the results section and were not discussed. Thus, they should be removed from the abstract.
Reply: Thanks for your suggestions. We have removed IL-23, IL-6, IL-22, Sclerostin, and Wnt1 from the abstract.
1.2. L29: doubled Dkk1. Which is the correct p value for Dkk1?
Reply: Thanks for your suggestions. We have revised the inappropriate presentation in the manuscript (Page 1, line 29), and corrected the erroneous data in the abstract.
- Introduction:
2.1. L58-59: IL-17 levels were shown to be elevated in RA patients. Revise the cytokine list for RA patients.
Reply: Thanks for your suggestions. We have revised the cytokine list for RA patients and added IL-17.
2.2. Unify abbreviations: IL17A or IL-17A, T-cells (L61) or T cells.
Reply: Thanks for your suggestions. We have revised and corrected them.
2.3. L82: revise the beginning of the sentence.
Reply: Thanks for your suggestions. We have revised and corrected the beginning of the sentence.
2.4. L92: it should be written ST instead of TS.
Reply: Thanks for your suggestions. We have revised and changed ST instead of TS.
- Materials and methods:
3.1. L119: delete "study".
Reply: Thanks for your suggestions. We have revised and deleted “study”.
3.2. L139-140: revise the sentence.
Reply: Thanks for your suggestions. We have revised and corrected the sentence.
- Results:
4.1. L187-188: the sentence states that 9 patients were treated with MTX but Table 1 shows that 8 patients were treated with MTX?
Reply: Thanks for your suggestions. We have revised and corrected to 8 patients.
4.2. L188-193: the described data is not presented with figure or table.
Reply: Thanks for your suggestions. These data have not been shown in new tables or figures but if allowed by the Editor we could include them as Supplementary Material.
4.3. Table 1: it is reasonable to show in separate column characteristics of patients treated with biologics after 5 years.
Reply: Thanks for your suggestions. If allowed by the Editor, we could include them as Supplementary Material.
4.4. Table 2: define the controls.
Reply: Thanks for your suggestions. We have added a definition of Controls in the footnote of Table 2.
4.5. Figure 1: revise the figure caption (... synovial tissue of patients treated with biologics or not.); indicate Patients instead of "Biologics", Without biologics instead of "Not" and With biologics instead of "Yes"; add IHC (%) on the y axis; define the type of statistical test used. What do the values 73, 1, 28, 43 indicate?
Reply: Thanks for your suggestions. We have revised the inappropriate presentation and corrected the Figure 1. Values 73, 1, 28, 43 indicate the coding number of patients with extreme values and we have eliminated them from the figure.
4.6. Table 3: what is the data for female patients?
Reply: Thanks for your suggestions. In Table 3 the coefficient for the variable "Male" is 2.059, suggesting that males are 7.84 times more likely to use biologicals compared to females, adjusted for other variables in the model.
4.7. Figure 2: improve the quality of the figure; define the controls and the statistical test used.
Reply: Thanks for your suggestions. We have changed the layout of the Figure to improve its visualisation.
4.8. L215, L 238: delete the comma after "constant" and "0.59".
Reply: Thanks for your suggestions. We have deleted comma after "constant" and "0.59".
- Discussion:
5.1. the section should include comparison with other IHC evaluations in RA and PsA patients and their possible relation with serum levels of the measured signaling molecules.
Reply: Thanks for your suggestions. We have revised some papers, but they did not include information about comparison with IHC evaluations in RA and PsA patients and their relationship with serum levels of the measured signalling molecules. We didn't find specific details on the correlation between serum levels and immunohistochemistry reactivity of IL-17A, TGF-β1, or Dkk1, in PsA or RA.
5.2. L286, L301: PAs or PsA?
Reply: Thanks for your suggestions. PAs has been changed to PsA.
5.3. L312: Double IL17A+CD8+.
Reply: Thanks for your suggestions. The duplicate has been removed.
5.4. L345: IHQ or IHC?
Reply: Thanks for your suggestions. IHC is correct. We have changed it.
- Conclusions: L369: delete "after 5 years".
Reply: Thanks for your suggestions. We have deleted "after 5 years".
We appreciate and benefit from your comments. Although some inherent limitations cannot be resolved, we have tried our best to modify them and are eager for your understanding. Thanks again.
Reviewer 3 Report
Comments and Suggestions for Authors
This is an interesting paper indicating the utility of synovial biopsy analysis in patients with active RA and PsA regarding stratification of therapy choice and treatment decisions. The authors suggest that specific biomarkers in synovial membrane may predict the necessity for the administration of biologic drugs in these conditions. I have a few comments which may improve the reading of the paper.
Introduction and Discussion sections are two long with significant degree of overlap. I suggest the shortening of both sections
In this regard a stronger rationale for the launch of the study is required
A more clinically oriented discussion will be very helpful as the message of the paper is somehow diluted amongst several findings
I believe a more extensive comments on the role of Dkk1 in joint inflammation characterizing RA (Mediterr J Rheumatol. 2017 Dec 22;28(4):174-182 )and bone remodeling/production charactering PsA and spondylarthritis (Mediterr J Rheumatol. 2022 Apr 15;33(Suppl 1):115-125.) should be included in the discussion to provide an overall explanation of the findings
Previous studies assessing the same question, namely biomarkers originating from synovial membrane biopsy should be discussed more extensively
Author Response
Reply to Reviewer 3
Comments and Suggestions for Authors
This is an interesting paper indicating the utility of synovial biopsy analysis in patients with active RA and PsA regarding stratification of therapy choice and treatment decisions. The authors suggest that specific biomarkers in synovial membrane may predict the necessity for the administration of biologic drugs in these conditions. I have a few comments which may improve the reading of the paper.
Reply: Thanks for your suggestions. We appreciate and benefit from your comments.
Introduction and Discussion sections are two long with significant degree of overlap. I suggest the shortening of both sections.
Reply: Thanks for your suggestions. We have shortened the Introduction and Discussion sections as much as possible.
In this regard a stronger rationale for the launch of the study is required.
Reply: Thanks for your suggestions. Initially, we set out to see what was happening with different osteoforming and osteodestructive cytokines in the synovial tissue of these patients, looking for something differential between the different pathologies. Subsequently, we followed these patients to see if any of the cytokines analysed could differentiate those patients with a milder clinical course of arthritis from those with a more moderate or severe evolution and who, therefore, required a more intense or aggressive therapeutic intervention.
A more clinically oriented discussion will be very helpful as the message of the paper is somehow diluted amongst several findings.
Reply: Thanks for your suggestions. Following your suggestion, we have added to the discussion a paragraph more oriented towards the clinical role of the findings.
I believe a more extensive comments on the role of Dkk1 in joint inflammation characterizing RA (Mediterr J Rheumatol. 2017 Dec 22;28(4):174-182 )and bone remodeling/production charactering PsA and spondylarthritis (Mediterr J Rheumatol. 2022 Apr 15;33(Suppl 1):115-125.) should be included in the discussion to provide an overall explanation of the findings.
Reply: Thanks for your suggestions. We have included a commentary in the Discussion using those references.
Previous studies assessing the same question, namely biomarkers originating from synovial membrane biopsy should be discussed more extensively.
Reply: Thanks for your suggestions. We have revised some papers, but they did not include information about comparison with IHC evaluations in RA and PsA patients and their relationship with serum levels of the measured signalling molecules. We didn't find specific details on the correlation between serum levels and immunohistochemistry reactivity of IL-17A, TGF-β1, or Dkk1, in PsA or RA.
We appreciate and benefit from your comments. Although some inherent limitations cannot be resolved, we have tried our best to modify them and are eager for your understanding. Thanks again.
Reviewer 4 Report
Comments and Suggestions for Authors
I completed the editing of a manuscript titled “Higher synovial immunohistochemistry of IL17A, Dkk1, and TGF-β1 reactivity in early psoriatic arthritis and in patients with rheumatoid arthritis could predict the use of biologics.” The structure of the manuscript (introduction, methodology, results, discussion) follows a logical sequence. . Introduction contains enough background informations. The study is well designed and conducted, but the following issues need to be clarified and completed:
1. There is an error in the abstract on line 29, Dkk1 is given twice with different p values. It should be corrected
2. Line 92 is an error, the abbreviation TS should be ST
3. Need to determine which patient samples (with which disease) were not suitable for quantitative IHC analysis?
4. The text shows that:
“At baseline, 9 patients were treated with MTX (61%), and no one patient was treated with biologics”
Table 1 shows that 8 (53.3%). Where do these discrepancies come from?
5. Table 1 does not explain the abbreviation r-axSpA
6. Table 2 shows the values for the control, but does not indicate whether there is statistical significance between the control and the study groups, i.e. PsA/RA with biologic and PsA/RA without biologics?
7. The description of Figure 1 does not indicate which patients it concerns - please add it
8. Figure 1 should include a legend.
9. Table 3 is unnecessary because it is a statistical table that causes confusion. It can be placed in Supplementary Materials
10. Descriptions in Figure 2 are poorly visible
11. Conclusions too far-reaching based on studies involving such a small number of patients, it should be added that they require further research.
Author Response
Reply to Reviewer 4
Comments and Suggestions for Authors
I completed the editing of a manuscript titled “Higher synovial immunohistochemistry of IL17A, Dkk1, and TGF-β1 reactivity in early psoriatic arthritis and in patients with rheumatoid arthritis could predict the use of biologics.” The structure of the manuscript (introduction, methodology, results, discussion) follows a logical sequence. . Introduction contains enough background informations. The study is well designed and conducted, but the following issues need to be clarified and completed:
- There is an error in the abstract on line 29, Dkk1 is given twice with different p values. It should be corrected
Reply: Thanks for your suggestions. We have revised the inappropriate presentation in the manuscript (Page 1, line 29), and corrected the erroneous data in the abstract.
- Line 92 is an error, the abbreviation TS should be ST
Reply: Thanks for your suggestions. We have revised and changed ST instead of TS.
- Need to determine which patient samples (with which disease) were not suitable for quantitative IHC analysis?
Reply: Thanks for your suggestions. This study included 13 patients diagnosed with psoriatic arthritis, 9 with RA, 18 with OA, and 4 with AS. Patients whose macroscopic or microscopic characteristics of the synovial membrane samples presented characteristics different from the underlying pathology (e.g., urate deposits or PPCD crystals, pigmented villonodular synovitis or vasculitis) that could interfere with the determinations were excluded, as well as poorly labeled or unlocatable samples.
- The text shows that:
“At baseline, 9 patients were treated with MTX (61%), and no one patient was treated with biologics”
Table 1 shows that 8 (53.3%). Where do these discrepancies come from?
Reply: Thanks for your suggestions. We have revised and corrected to 8 patients.
- Table 1 does not explain the abbreviation r-axSpA
Reply: Thanks for your suggestions. We have revised and explained the abbreviation r-axSpA. Radiographic axial spondyloarthritis (r-axSpA) is mandatorily defined by evident radiographic structural damage.
- Table 2 shows the values for the control, but does not indicate whether there is statistical significance between the control and the study groups, i.e.PsA/RA with biologic and PsA/RA without biologics?
Reply: Thanks for your suggestions. The p<0.05 obtained in the comparative analysis of the 3 groups have been considered. We have added it to the footnote to Table 2 and changed Wilcoxon’s signed rank test to Kruskal-Wallis’s test.
- The description of Figure 1 does not indicate which patients it concerns - please add it
Reply: Thanks for your suggestions. Table 2 and Figure 1 express the same issue, providing a more complete picture. Figure 1 shows the IHC expression of the 3 cytokines in all patients divided into 2 groups: those who have received biological treatment and those who have not received biological treatment. We have added it to the footnote.
- Figure 1 should include a legend.
Reply: Thanks for your suggestions. We have added a footnote to Figure 1.
- Table 3 is unnecessary because it is a statistical table that causes confusion. It can be placed in Supplementary Materials
Reply: Thanks for your suggestions. If allowed by the Editor, we could include them as Supplementary Material.
- Descriptions in Figure 2 are poorly visible
Reply: Thanks for your suggestions. We have changed the layout of the Figure to improve its visualisation.
- Conclusions too far-reaching based on studies involving such a small number of patients, it should be added that they require further research.
Reply: Thanks for your suggestions. We have changed the conclusions in a less far-reaching sense, adding that more future research is needed on this topic.
We appreciate and benefit from your comments. Although some inherent limitations cannot be resolved, we have tried our best to modify them and are eager for your understanding. Thanks again.
Round 2
Reviewer 1 Report
Comments and Suggestions for Authors
The authors have done their best to improve the article both in terms of writing and presentation. However, these improvements are only cosmetic and do not address the fundamental problem of the very small size of the patient cohorts used in this study.
Author Response
Reply to Reviewer 1
Comments and Suggestions for Authors
The authors have done their best to improve the article both in terms of writing and presentation. However, these improvements are only cosmetic and do not address the fundamental problem of the very small size of the patient cohorts used in this study.
Reply: Thanks for your comments. As you know, patient recruitment for such studies is extremely complicated. It involves getting patients in the early stages of the disease, who have persistent inflammation in the knee joint that has not improved with conventional first-line treatments, and who have not started biologic treatment in the case of inflammatory arthropathies such as rheumatoid arthritis or psoriatic arthritis. As a result, publications are scarce and sample sizes are small. It is recommended to include between 30 and 50 participants, who should possess the attributes to be measured in the target population. (Earl R. Babbie. The Practice of Social Research. 15th Edition. Cengage AU, 2020). When the proportion of eligible subjects who will refuse to participate or provide inadequate information will be unknown at the beginning of the study, approximate estimates are often possible using information from similar studies in comparable populations or an appropriate pilot study (Whitley E, Ball J. Statistics review 4: Sample size calculations. Crit Care. 2002; 6:335–41). We deeply regret that we cannot provide a larger sample size, but we intend to continue to increase the sample size in future projects.
We appreciate and benefit from your comments. Although some inherent limitations cannot be resolved, we have tried our best to modify them and are eager for your understanding. Thanks again.
Reviewer 2 Report
Comments and Suggestions for Authors
The manuscript has been significantly improved.
However, there are several points that need to be further addressed:
1. L17: Please revise the beginning of the sentence. The objective of the study was to examine synovial immunohistochemistry reactivity of cytokines related to inflammation ...
2. L57: insert citation.
3. Look at L223-224 and L228-229. The sentences report the same result but with different p value. Why?
4. Figure quality should be improved. Figure captions should be presented in accordance to the journal guidelines.
5. The results from the ROC curve analyses should be better discussed.
Author Response
Reply to Reviewer 2
General comment:
Comments and Suggestions for Authors
The manuscript has been significantly improved.
However, there are several points that need to be further addressed:
- L17: Please revise the beginning of the sentence. The objective of the study was to examine synovial immunohistochemistry reactivity of cytokines related to inflammation ...
Reply: Thanks for your comments. Following your suggestion, we have changed it to "The study aimed to investigate the immunohistochemical reactivity of synovial cytokines associated with inflammation..."
- L57: insert citation.
Reply: Thanks for your comments. Following your suggestion, we have inserted the citation [6].
- Look at L223-224 and L228-229. The sentences report the same result but with different p value. Why?
Reply: Thanks for your comments. As you rightly point out, statistical significance is different. IL-17A IHC reactivity was higher in males, both in the bivariate analysis (p 0.02) and in the regression analysis (p 0.013).
- Figure quality should be improved. Figure captions should be presented in accordance to the journal guidelines.
Reply: Thanks for your comments. Following your suggestion we have improved the quality of the figures and presented the figure captions following the guidelines of the journal.
- The results from the ROC curve analyses should be better discussed.
Reply: Thanks for your comments. Following your suggestions we have further discussed the results of the ROC curve analysis.
We appreciate and benefit from your comments.
Reviewer 3 Report
Comments and Suggestions for Authors
The authors have addressed the comments
Author Response
Reply to Reviewer 3
Comments and Suggestions for Authors
The authors have addressed the comments
Reply: Thank you very much for your kind comments on our manuscript.
Reviewer 4 Report
Comments and Suggestions for Authors
I ask the authors to clarify two more points.
1. The authors write: " Forty were valid for IHC quantification analysis, and four were not valid for the study". Did the 4 samples that were eliminated from the study belong to RA, PsA, OA or AS patients?
2. A statistically significant result of the Kruskal-Wallis test only tells us that at least one of the groups is different from another group. Therefore, we then perform post-hoc tests (usually the Dunn or Dunn test with Bonferroni correction) to check exactly which groups differ from each other. What post-hoc test did the authors use? Was there statistical significance between the control group and PsA/RA with biologics or the control group and PsA/RA without biologics? - should be completed in the table 1.
Author Response
Reply to Reviewer 4
Comments and Suggestions for Authors
I ask the authors to clarify two more points.
- The authors write: " Forty were valid for IHC quantification analysis, and four were not valid for the study". Did the 4 samples that were eliminated from the study belong to RA, PsA, OA or AS patients?
Reply: Thanks for your comments. Two patients with arthritis (one with PsA and one with RA) and two with osteoarthritis were excluded because the IHC reading of BMP4 and Wnt5b could not be performed correctly. We have added this explanation to the manuscript. However, the IHC reactivity of Dkk1, BMP2, TGF-b1, and IL-17A in the ST did not vary, even taking these patients into account.
- A statistically significant result of the Kruskal-Wallis test only tells us that at least one of the groups is different from another group. Therefore, we then perform post-hoc tests (usually the Dunn or Dunn test with Bonferroni correction) to check exactly which groups differ from each other. What post-hoc test did the authors use? Was there statistical significance between the control group and PsA/RA with biologics or the control group and PsA/RA without biologics? - should be completed in the table 1.
Reply: Thanks for your comments. We have followed his suggestion and improved Figure 2 by adding the pairwise statistical significance automatically obtained by the Dunn-Bonferroni test when the Kruskal-Wallis test was calculated. We have also added an explanatory commentary at the bottom of the figure and in the Statistical Analysis section in the Material and Methods section.
We appreciate and benefit from your comments.